# COVID-19 Experience Transforming the Protective Environment of Office Buildings and Spaces

**Panupant Phapant** [1] , **Abhishek Dutta** [2] **and Orathai Chavalparit** [2],*

[1] Interdisciplinary Program of Environment, Development and Sustainability, Graduate School, Chulalongkorn University, Bangkok 10330, Thailand; panupant.p@gmail.com

[2] Department of Environmental Engineering, Faculty of Engineering, Chulalongkorn University, Bangkok 10330, Thailand; duttabob@gmail.com

* Correspondence: orathai.c@chula.ac.th; Tel.: +66-815-536-884

**Abstract:** The COVID-19 pandemic has affected human life in every possible way and, alongside this, the need has been felt that office buildings and workplaces must have protective and preventive layers against COVID-19 transmission so that a smooth transition from 'work from home' to 'work from office' is possible. However, a comprehensive understanding of how the protective environment can be built around office buildings and workspaces, based on the year-long experience of living with COVID-19, is largely absent. The present study reviews international agency regulation, country regulation, updated journal articles, etc., to critically understand lessons learned from the COVID-19 pandemic and evaluate the expected changes in sustainability requirements of office buildings and workplaces. The built environment, control environment, and regulatory environment around office buildings and workplaces have been put under test on safety grounds during the pandemic. Workers switched over to safely work from home. Our findings bring out the changes required to be affected in the three broad environmental dimensions to limit their vulnerability status experienced during the pandemic. Office building designs should be fundamentally oriented to provide certain safety protective measures to the workers, such as touch-free technologies, open working layouts, and workplace flexibilities to diminish the probability of getting infected. Engineering and administrative control mechanisms should work in a complementary way to eliminate the risk of disease spread. Country regulation, agency regulations, and operational guidelines need to bring behavioral changes required to protect workers from the COVID-19 pandemic.

**Keywords:** COVID-19; office buildings and workplaces; office-built environment; control environment; regulatory environment

## 1. Introduction

The COVID-19 outbreak is one of the most significant global pandemics in human history. As the World Health Organization (WHO) recently announced, the COVID-19 pandemic is slowly becoming an inescapable part of human life and work [1,2]. WHO data revealed that, as of September 2021, more than two hundred and thirty million people had been infected by and almost five million have died from COVID-19 [3]. Even as humans are looking to minimize COVID-19's impact on daily life through specific vaccine availabilities, different evolving virus variants have held the human world in the doldrums. This pandemic has created significant challenges in public health and other various areas, including politics, economics, education, and social behavior. The outbreak has resulted in a global economic decline, and economic growth in every global region was forecasted to decrease significantly. For instance, the World Bank predicted the economic growth of Asia and the Pacific to be a mere 0.5%, and South Asia's economy to contract by 2.7%, Europe and Central Asia's by 4.7%, and Latin America's by 7.2% compared to the onset of the pandemic [4]. Even after a year of struggling to cope with COVID-19 and

vaccine introductions, the International Monetary Fund's outlook for the world's economy indicates very high uncertainty and uneven economic recovery during 2021 [5].

The modes of transmission for the SARS-CoV-2 (COVID-19 causing) virus are similar to those of the flu, SARS-CoV-1, MERS, and influenza viruses, which include contact, droplet, airborne, fomite, fecal–oral, blood borne, mother-to-child, and animal-to-human transmission [6]. Pamidimukkala et al. [7] concluded that the virus can be transmitted from person-to-person and causes symptoms that include fever, dry cough, fatigue, and shortness of breath. Due to its strong infectious nature, extended incubation period, difficulty in detection, and vagueness in transmission modes, COVID-19 is becoming a very difficult disease to control [8]. The pandemic has resulted in the enactment of social distancing policies. It has dramatically affected many workforces and workplaces as governments worldwide have implemented various social restrictions to curb infection rates, such as closing schools and asking people to work from home [9]. Many companies have been forced to shut down, affecting many countries' economies, while employees whose jobs have been retained have experienced heightened concerns for their mental health and physical wellbeing. People worldwide are panicking about the virus's fast transmission rates and are isolating themselves from office buildings, and a significantly large percentage of every country's population is spending their time indoors only. Nevertheless, after months of strict quarantine, a reopening of society is inevitable. Many countries are planning exit strategies to progressively lift lockdown measures without leading to an increase in the number of COVID-19 cases [10]. Due to the virus's ongoing persistence, engineers and scientists are trying their best to innovate and discover new cures to fight against this virus [11–14]. Whether official or residential buildings are considered, they should all be equipped with multiple protective layers to contain the spread of viruses, to the point where current pandemics will slowly disappear [15]. By implementing multiple protective layers against viruses, humans will create an environment in which the transmission and mutation of fast-adapting viruses will be minimized.

COVID-19 has affected human life in every possible way; in turn, human patterns that are changing because of COVID-19 could lead to insights that are useful in the fight against COVID-19 [16]. Ever since COVID-19 surfaced, people have started deliberately altering building infrastructures, house interiors, and lifestyles in ways to provide some protective layers against the virus [17]. Fear of COVID-19 has redirected customers' preferences from hypermarkets to decentralized commercial places for quicker shopping. Decentralization, redistribution, and restructuring have become key design concerns since COVID-19. Even the vision of cities' futures has changed in the minds of city dwellers, who are now favoring six-lane cycling tracts over six-lane highways as their favorite option of commuting to and from offices and schools [18].

In their review paper, Honey-Roses et al. (2020) [19] concluded that the long battle with COVID-19 will make the world rethink the design of urban public spaces to serve people's best interests. Sepe (2021) [20] went one step further and pointed out that the COVID-19 pandemic has raised questions about the validity of the principles and practices of the 'Charter of Public Space' being followed by several countries when designing public spaces. Dogan et al. (2020) [21] indicated a positive correlation between meteorological variables, and ambient air quality ($PM_{2.5}$, $PM_{10}$) with COVID-19 cases, while Shahzad et al. (2021) [22] showed that air quality and temperature positively correlated with COVID-19 death cases. Emmanuel et al. (2020) [23] reemphasized the importance of introducing fresh air into crowded and poorly ventilated buildings [7].

## 2. Workplace Regulations to Mitigate COVID-19

Since workers can come into close contact with colleagues and visitors coming from different localities with various COVID-19 pandemic infectivity levels, virus exposure can occur at any time within the workplace [24]. Therefore, regulatory guidelines and directives are essential in fighting the COVID-19 pandemic in workplace settings. Policies aimed at protecting workers also prevent community transmission of the virus and protect

national economies by maintaining open and safe workplaces [25]. Apart from international health regulations, countries also need to implement pandemic control regulations and directives that reflect their specific needs and legislative backgrounds. Sector-specific guidelines can give further clarity to business operators and benefit different sectors of the economy. Both governments and industry associations must actively support devising industry-specific risk and safety guidelines that workers need to follow. Tokazhanov et al. (2020) [26] noted that existing building sustainability ratings based on "environmental impact" and "energy performance" required total overhauling as the current pandemic has exposed their inadequacies. Jiang et al. (2021) [27] pointed out that drastic changes in the spatial distribution of living spaces will be required to control the spread of the epidemic in the community. The Architectural Society of China (2020) [28] emphasized the need for improving buildings' health performances as buildings are inevitably linked to their communities in epidemic prevention and control measures. Awada et al. (2021) [29] confirmed that the current layout of buildings cannot effectively withstand the spread of COVID-19 and therefore require changes. Valizadeh et al. (2021) [30] further suggested that alteration of the indoor environment is required as most infections occur indoors.

The COVID-19 pandemic has forced businesses to allow officials to stay home and remain active for official engagements through the online presence. Experts are apprehensive that working from home will not be productive as workers are away from their coworkers and managers [31]. Additionally, studies have shown that workers cannot disengage from official work in time when working from home, which has led to increased worker stress and decreased productivity [32]. In fact, both managers and workers are unhappy with the continuance of working from home. Currently, many countries are in the middle of their second or third waves of the pandemic, and yet both government and private establishments are already planning for the eventual reopening of offices to employees. Consequently, the persistence of the coronavirus pandemic has led to reconsiderations of existing office buildings and official workplaces; both office buildings and workplaces need to become more resilient against COVID-19 disease spread for the eventual return of workers. Companies are preparing for the return of employees to their offices with extensive safety arrangements to prevent COVID-19 transmission. At this juncture, it is essential to review how office buildings and workplaces can build up protective and preventive layers against COVID-19 transmission among employees and workers.

In that context, by the literature review, this paper intends to address the following questions: (1) what are the deficiencies in office buildings and their layouts in regards to the infection spreading propensity during the COVID-19 pandemic; (2) to what extent do COVID-19 regulations and guidelines bring about safety to officials; and (3) what are some emerging solutions that will make workplaces safer against COVID-19 when considering office building structural designs and workplace layouts, the virus spread elimination, capacity control, and effective environmental regulations? The answer to these questions will be essential for policymakers for managing a smooth transition from 'work from home' to 'work from the office' as usual.

## 3. Methodology

COVID-19 related scientific literature grew much faster than any other pandemic the world has witnessed so far. This paper gathered data and information through reviewing the literature. The literature review took place during June–July 2020 and focused mainly on the Web of Science (WOS), Google Scholar, and PubMed databases for journal paper search. The key words used for searching were "office environment and COVID-19", "office regulations and COVID-19", "office layouts and COVID-19", and "office safety and COVID-19". A total of 112 journal papers were retrieved using this method. On reviewing the above journal papers, 19 articles were excluded because of content similarity or they dealt with other aspects of the COVID-19 outbreak not covered by the objectives of the present study. The current article is a narrative review of the 112 papers related to the

protective and preventive office environment relevant to the COVID-19 pandemic. This study also used important information provided by credible blog sites.

## 4. Measure/Regulation to Contain the Spread of the COVID-19 Disease in Offices

SARS-CoV-2, the virus that causes COVID-19, attacks the respiratory tract of infected individuals, causing breathing troubles, fevers, and coughs. Infection with this virus primarily causes respiratory illness that ranges from no symptoms, mild to severe symptoms, or even death. According to current knowledge about SARS-CoV-2, transmission primarily occurs with close contact with an infected person or coughing and sneezing. Viral transmission rates depend on the amount of viable virus being shed and expelled by the infected person, the types of contact an infected person has with others, the settings where exposure occurs, and what preventative measures are in place [25]. Similar to regular flu viruses, SARS-CoV-2 droplets are released by infected persons through coughing or exhalation and spread by touch or inhalation. Touch transmission occurs when individuals touch contaminated surfaces and then their face, nose, eyes, or mouth, while airborne transmission occurs when individuals merely breathe in air that contains viral droplets. Indoor climate-controlled environments exacerbate airborne viral transmissions, especially when ventilation systems are malfunctioning or simply inadequate [33]. As such, strategies to mitigate airborne transmission, such as de-densification, will not be adequate to contain virus spread. At the end of the day, managers and employees will need foolproof, tailor-made COVID-19 protocols that only take the regulations and guidelines produced from the international and national levels as basic considerations in order to protect their wellbeing during this pandemic emergency.

### 4.1. Agency Regulations

International organizations/agencies, such as the WHO and International Labour Organization (ILO), have released various health regulations to counter the spread of COVID-19 disease in workplaces. These regulations have emphasized cleaning working surfaces, maintaining good hygiene, regular hand washing, promoting good respiratory hygiene, and developing well-defined meeting protocols in workplaces [34]. The WHO [25,34–38] and ILO [39–41] have also provided national and local governments, employers, workers and their representatives, and occupational health services with practical guidance on preventing COVID-19 outbreaks at work by minimizing exposure to and transmission of SARS-CoV-2 among workers [25]. The USA Occupational Safety and Health Administration (OSHA) [42,43] also published regulations related to infection prevention and industrial hygiene practices for the workplace. OSHA regulations emphasize employers' responsibility to provide employees with a workplace free from recognized hazards that can cause death or serious physical harm [43]. In response to the COVID-19 pandemic, the Centers for Disease Control and Prevention (CDC) [44–47] issued updated employer regulations to ensure safe and healthy workplaces. The CDC made it compulsory for employers to regularly assess workplace hazards and implement an effective hazards control system [44]. Similarly, the European Centre for Disease Prevention and Control (ECDC) [48–51] recommended the installation of heating, ventilation, and air conditioning systems (HVAC) for minimizing airborne transmission of COVID-19 in workplaces and measures which reduce the risk of airborne SARS-CoV-2 transmission [52]. Additionally, the American Society of Heating, Refrigerating and Air-Conditioning Engineers (ASHRAE) recommends various strategies to fight against COVID-19 in workplaces, especially ventilation improvement [53–58].

Supplementary Table S1 shows a summary of the preventive and protective recommendations against COVID-19 established by various international organizations. The hierarchy of controls—that is, the levels of risk management—comprises of five groups which include, in order from most to least effective, elimination, substitution, engineering control, administrative and organization control, and personal protective equipment (PPE). Elimination is the most effective strategy; however, it is not always possible to completely

eliminate hazards in any operations, and therefore a combination of the other measures is required. Engineering controls involve considering the details of building facilities and equipment that can be improved to reduce exposure to hazards without reliance on worker behavior.

*4.2. Country Regulations*

Besides workplace guidelines from various international organizations, different countries have also presented their own regulations and measures to contain the spread of the COVID-19 disease in offices and workplaces. Although a country's success in containing the pandemic is not the outcome of solely effective workplace regulations, successful pandemic containment is almost impossible without the latter. The Bloomberg COVID-19 Resilience Ranking provides a serviceable snapshot of COVID-19 prevention successes of 53 countries around the world. In the ranking, countries listed with effective regulations vis-à-vis achieved success in containing COVID-19 were New Zealand, Singapore, Australia, Israel, South Korea, Finland, Norway, Denmark, Mainland China, and Hong Kong [59]. The examples of various country regulations are as follows:

Singapore [60,61]: Work-from-home became the default mode of work in this country during the COVID-19 pandemic; penalties were implemented for anyone violating this policy. Employers were instructed to take care of their workers and workplaces, especially individuals who became unwell in the workplace. To fulfill this, employers were instructed to implement safe management measures such as monitoring plans and safe management officers. Personal hygiene, safe distancing, and mental health measures were also essential strategies in this country's success in containing the pandemic.

South Korea [62]: The government of the Republic of Korea launched the "All about Korea's Response to COVID-19" initiative to help employers and employees maintain good hygiene and social distancing in the workplace. Strategies included avoiding crowded work environments and implementing work-from-home and flexible work schedules.

Finland, Norway, and Denmark: These Nordic countries all had high rankings for best COVID-19 management by implementing advisories for employers and employees [63–66], recommending remote work [67], and re-opening offices [68].

China: The State Council for the People's Republic of China published guidelines on staying safe from COVID-19 in workplaces for both employers and employees. Strategies included [69] guidelines on the commute to work, using stairs (avoiding elevators) upon arrival at work, guidelines in the office, and guidelines for after work. Holiday practices were also outlined [70].

Hong Kong: The Centre for Health Protection, Department of Health, advised for the prevention of COVID-19 in the workplace and businesses (Interim) [71] by maintaining good personal and environmental hygiene in workplaces, maintaining toilet hygiene, and maintaining the hygiene of overnight rooms

*4.3. Operational Guidelines*

The risk of infection spread across employees in a workplace also depends on the characteristics of the working environment and workplace operational guidelines. For example, Dyal et al. (2020) [72] reported the ordeal of many workers in the meat processing sector when group infections occurred in this sector's closed factory environments which often lack adequate ventilation. Agostinho (2020) [73] pointed out that COVID-19 spread quickly in the meat processing sector because of its intrinsic processing requirement which prevented coworkers from maintaining a six-foot safe working distance between one another. Furthermore, Van Den Berg et al. (2021) [74] emphasized the importance of maintaining physical distancing between 3 versus 6 feet in public schools with mandatory face mark wearing. The data reveals that lower distancing can be applied in a school setting. Therefore, Ijaz et al. (2021) [75] emphasized the necessity of framing policies and working protocols exclusively for meat production and supply chain to curtail disease spread.

Elsewhere, health sector workers were the most vulnerable to COVID-19 during the pandemic's peak periods in some respective countries due to the sector's intrinsic work patterns [76]. Iavicoli et al. (2021) [77] presented a general workplace risk assessment framework based on three intrinsic requirements: exposure, proximity, and aggregation. Unsurprisingly, health and social workers had higher risk infectivity based on this assessment than workers in education, arts, entertainment, and recreation. This fact supports the notion that workers in different sectors cannot be effectively protected from the COVID-19 pandemic by a single safety guideline or protocol [78]. Instead, the understanding of sector-specific risk factors and workplace characteristics must be a prerequisite for developing effective, tailor-made safety guidelines and protocols for a particular workplace. Finally, workers must receive timely communications about updated guidelines and protocols, such that strategies are correctly followed and problematic misperceptions between managers and workers can be avoided [79].

## 5. Effects of COVID-19 Pandemic on Office and Workplace Sustainability

In the long duration of the COVID-19 pandemic, since its outbreak, the pandemic has provided varied office working experiences that have given a greater understanding of workplace vulnerabilities against disease transmission. COVID-19 has affected the working style of professionals, irrespective of whether they worked from home or in commercial office spaces. In some industrial operations, where working from home was not possible, business owners and facility managers had to make arrangements and implement engineering and administrative measures to ensure the safety of on-site work. Workplace layouts were adjusted to provide some distance between neighboring workstations and ventilation systems were improved. Administrative instructions were issued to employees to maintain distancing requirements from other colleagues while at the office. All these measures acted as collaborative protective layers against the spread of COVID-19 to make office buildings and workplaces coherent working places once again.

### 5.1. Engineering Control Environment in Workplaces

Many workers converted their homes to makeshift offices during the COVID-19 pandemic. Working from home has given workers flexibility in performing official work and, more importantly, protection from virus infection. As managers and workers have become more experienced in handling their own safety during the pandemic, their desire to return to regular office-working arrangements also increased. However, with the dangers of the virus still present, office workplaces need to be modified to provide workers with a sense of security such that their normal productivity could resume [80–82]. The dangers of COVID-19 should compel office managers to develop the built environment around offices with protective layers to prevent infection spread among workers. In this context, experiences of living with COVID-19 have provided many practical insights.

Workplaces are vulnerable areas where rapid multiplication of COVID-19 cases or infection clustering is possible. Although allowing employees to work from home can negate the hazards of the workplace, most workforces include a large portion of employees that cannot work from home due to the criticality of their sectors. On top of that, whether workers work in a factory or plush office chamber, their exposure to infection hazards also put their family members and coworkers at risk [83]. Considering how infected workers can quickly spread pandemic viruses to their urban residential centers, the workplace must have protective layers to protect workers from virus transmission [84].

Günther (2020) [85] reported on the evidence of how a lack of infection control efforts from management in a German food processing factory led to the continuous exposure of workers to recirculated infected air, resulting in a rapid increase in new COVID-19 cases. Elsewhere, Herstein et al. (2021) [86] reported on a similar super-spreading event of COVID-19 in a meat processing factory in Nebraska, USA, once again due to a lack of infection controls. Investigations into these rapid virus transmission cases in factories revealed the absence of three important infection control measures: lack of fresh air cir-

culation in a closed space, non-existent physical distancing between coworkers during working situations, and lack of administrative controls. It is essential for management to develop well-thought-out infection control plans to protect workers from the COVID-19 disease, especially where risk propensity is inherently high, such as open office space environments [87]. Management's advanced preparedness to prevent COVID-19 and other pandemic diseases will be very relevant for their workers' safety. Infection prevention should take top priority in any manager's plans against COVID-19. Successful infection prevention hinges on correctly assessing the varying degrees of risk relevant to different industrial and office setups and managing these risks with effective control strategies [88]. Managers will be challenged to understand the infection-spreading gateways in their respective controlled areas and then implement the right tools and techniques to close these gateways. In the case of the earlier-mentioned meat factories, workers commented that managers allowed for closed spaces with reduced fresh air circulation, leading to the rapid spread of infection [89]. Workers also noted that the virus's longer surviving ability in a low-temperature closed room was not considered before allowing workers to be present [85]. Employees are demanding to return to offices that are more adaptive against COVID-19. Adaptive office facilities may include flexible desk arrangements, improved touch-free technologies, less crowding made possible by employee roster presence plans, frequent sanitization, and flexibility of switching to remote working arrangements as needed [90].

Building owners and policymakers must collaborate with the office manager to implement the infection protective and preventive strategies by following agency regulations and country measures. Furthermore, they have to take control actions against COVID-19, which may include (a) implementing engineering controls (such as ventilation) to prevent airborne virus transmission and (b) implementing administrative controls and COVID-19 protocols that apply to everyone present in the workplace.

### 5.1.1. Ventilation

Literature on COVID-19 supports airborne transmission of the SARS-CoV-2 virus, mainly by respiratory droplets smaller than 5 microns. However, although larger respiratory droplets (i.e., more than 10 microns) cannot be suspended in the air for much time, they can still cause disease transmission when they accumulate on and contaminate surfaces. There is some contradicting opinion regarding the duration of aerosol viruses staying active while afloat, how fresh air dilution affects viral loads, and how far viruses can travel while airborne. However, some existing case studies undeniably confirm the risk of airborne COVID-19 transmission. Aerosol SARS-CoV-2 viruses are a genuine concern for workers in closed office spaces. Evidence has pointed out that aerosol viruses can remain active in the air for some time and make a direct airborne journey into workers' respiratory tracts [91].

Research from different countries during the first COVID-19 waves highlighted the possible respiratory spread of the novel coronavirus under low temperature and low relative humidity (RH) conditions [92,93]. Low RH and temperature environments supported the rapid transmission of the novel coronavirus across the population [94]. Management's responsibility will be to implement engineering controls that prevent the spread of COVID-19 in workplaces, be it by installing or maintaining adequate air replenishing systems or climate control systems. In cases where engineered measures are inadequate or impractical to install, managers must look into other ways to prevent disease transmission in their workplaces.

### 5.1.2. Distancing and Building Layout

The built environment around offices needs to be reshaped and adapted under the backdrop of the COVID-19 pandemic [95]. The pandemic has shown that conventional office design concepts have been too narrowly focused on productivity and economic efficiency. Although high-rise commercial buildings can accommodate huge worker popu-

lations, they also expose more people to a higher risk of contamination through human contact. Interior office spaces that opted for less square footage per worker for the sake of cost-effectiveness have inversely become the least desirable option under COVID-19 [96,97]. State-of-the-art high-speed elevators in sophisticated multistory office buildings have suddenly become places of high infection spread propensity; employees now perceive offices with sprawling staircases as safer and more beneficial to their health [98].

The attitude of office managers towards open office plans changed fast when they had to reduce virus propagation among office workers during COVID-19 [80]. Open office layouts can crowd and confine many employees into a shared space, thereby facilitating aerial viral transmission when infected coworkers cough and expel coronavirus droplets. The SARS-CoV-2 virus can be present in aerosols for up to three hours, during which time people can be infected [99,100]. As society develops an in-depth understanding of the plausible modes of transmission for the SARS-CoV-2 viruses, more and more employees find existing open space offices unacceptable [101]. Literature on COVID-19 also supports single-occupancy office rooms and even suggests keeping rooms empty for some time, depending on airflow rate and room size, before new employees enter [102].

### 5.1.3. Facilities and Equipment

Of course, devices with touch-free technology can altogether eliminate physical contact by workers. Demand to make day-to-day office operations touch-free will be increasing [103]. Before the COVID-19 pandemic, employees had to touch many facilities to operate them. Day-to-day office work involved touching devices such as biometric fingerprint scanners, computers, printers, tea and coffee vending machines, and many more. With the coronavirus situation, employees have become more reluctant to operate office facilities via touch. After COVID-19, there will be a demand for offices where employees rarely need to touch items with their hands. At the same time, lifts and coffee vending machines can be operated using personal smartphones [104]. Offices will need to be equipped with touch-free technologies as employees return to the workplace [105]. Management must plan to use affordable, available, and workable technologies to drastically reduce office touchpoints to allay workers' fears of infection. Otherwise, workers may become hesitant in regularly attending offices that overly rely on touch-based operations.

Individuals can be infected with COVID-19 by touching different virus-contaminated surfaces in the office. Van Doremalen et al. (2020) [106] indicated that the active period of SARS-CoV-2 viruses vary based on the type of surface they are present on. SARS-CoV-2 remains active comparatively longer (2 to 3 days in laboratory condition) on plastic and stainless-steel surfaces but much shorter (1 day) on cardboard surfaces. Copper surfaces, despite being metal, provide the minimum life of viruses. Laboratory experiments have also shown that a single contaminated door handle can cause rapid multiplication of infected persons in an office within hours [107]. Surfaces that can retain viruses longer need to become less frequent touch points. Workers need to be aware of the risk profile of their immediate office environment and become more strategic in keeping infection possibilities at a low level [108].

### 5.1.4. Water and Sanitary System Safety

During the temporary building closures due to COVID-19, the water tank of the buildings are likely to have microbiological contaminants [109], which may take a long period to recover to the normal levels. Furthermore, increased legionella [110] can cause health hazards for the building's occupants.

### 5.2. *Administrative and Organizational Control Environment in Workplaces*

It is of particular importance for office employees to be well aware of COVID-19 safety precautions such as mask-wearing. It has been found that the doffing of COVID-19 protective equipment is common in workplaces, which invites infection risks to all who are

present [111]. Additionally, workplace visitors must also be delicately managed to prevent external virus sources from penetrating the workplace's disease-prevention layers.

To effectively combat COVID-19, every employee in a typical workplace must be unitedly involved in following protocols designed to prevent disease transmission. Lapses or suppression of disease symptoms by a single worker can put entire workplace populations at risk of infection that may lead to the closure of the entire office, depending on local statutory guidelines [112]. As such, office administrators must frequently and effectively communicate basic COVID-19 prevention guidelines to all employees. More importantly, they must take on the challenge of maintaining employees' motivation in adhering to enforced guidelines and systems [113].

### 5.3. Vulnerabilities of Office Buildings and Workplaces

For workers to return to their respective workplaces, different pandemic-resilient requirements must be ingrained into the collective system of building architectures, layouts, safety measures, and many more. If these much-required protective layers are successfully executed in office and factory environments, they can ensure the sustainability of workplaces during the pandemic. The key vulnerabilities in the implementation of preventive COVID-19 regulations, engineering control, and administrative control have been identified and summarized in Table 1. Each vulnerability needs to be addressed to minimize disease transmission risks among employees and built antivirus protective layers.

**Table 1.** Vulnerabilities of office buildings and workplaces based on experiences with COVID-19.

| Category | Subcategory | Vulnerabilities | References |
|---|---|---|---|
| Regulatory Environment | International Agency Regulations | • Virus exposure can occur anytime at workplaces where no regulations can help.<br>• Various health regulations from the various international agencies to be followed in workplaces are unknown.<br>• A simple translation of the requirements documented at the international, national, and sectoral levels to office-specific guidelines and protocols may not be adequate. | [24,114,115] |
| | Country Regulations | • National level COVID-19 guidelines may have some mismatches with international guidelines.<br>• Country regulations may reflect national COVID-19 resiliency but are not office specific.<br>• Country regulations only serve as general reflections of COVID-19 threats. | [59,115] |
| | Operational Guidelines | • Guidelines are rapidly changing and evolving, which may lead to confusion.<br>• Workers may not receive updated COVID-19 operating procedures and guidelines in a timely fashion. | [77–79,116] |
| Engineering Controls Environment | Ventilations | • Filtered air or fresh air supplies are not available in some offices and workplaces.<br>• Lack of infection controls through engineering means in office spaces.<br>• Management may not know proper indoor temperature and humidity control protocols that make the COVID-19 disease less infectious. | [86,91,92,94] |

**Table 1.** *Cont.*

| Category | Subcategory | Vulnerabilities | References |
|---|---|---|---|
| Engineering Controls Environment | Distancing and Building Layout | • Indoor offices are rigid and unadaptable structures that prohibit workers from maintaining a safe distance from one another.<br>• Existing open plan office interior designs deemed no longer appropriate due to high COVID-19 transmission propensity.<br>• High-rise commercial buildings have more risk of contamination. | [15,80,96–98,100] |
| | Facilities and Equipment | • Day-to-day work requires touching of different gadgets and surfaces, increasing the chance of virus transmission.<br>• Hands-free cultures are generally absent and difficult to introduce into traditional offices.<br>• Automated office environments with less physically operated systems are absent.<br>• Prevailing work cultures are not resilient in preventing the spread of infectious diseases such as COVID-19.<br>• Historically, office equipment and facilities were not designed for preventing infectious diseases.<br>• SARS-CoV-2 has been discovered on surfaces in the isolation area where suspected patients were being held. | [108,117–119] |
| | Water and Sanitary System Safety | • Stagnant water can lead to poor water quality, notably microbiology and legionella, which can harm building occupants. | [109,110] |
| Administrative and Organizational Controls Environment | Workplace Safety Measures | • Employees doffing COVID-19 protective gear in the workplace enhances the risk for colleagues. | [111] |
| | Cleaning and Disinfection | • Typical office space cleaning policies and procedures are inadequate at preventing infection risks. | [88] |
| | Emergency Plan Development | • Office surveillance systems based on staff attendance are almost useless during the COVID-19 pandemic and require upgrades.<br>• Employees suppressing disease symptoms create risk for entire offices. | [86,112] |
| | Business Continuity Guideline | • Management is not adequately assessing the carrying degrees of risk relevant to different industrial and office setups. | [88] |
| | Education and Communication | • Less motivated employees may avoid being trained on COVID-19 protocols or following the protocols. | [113] |

## 6. Protective and Preventive Layers for Office Buildings and Workplaces

The current pandemic has shown the need for the built environment to be unfriendly towards severe acute respiratory syndrome coronavirus 2 (SARS-CoV-2) types of viruses [15]. A building's resiliency against the coronavirus will become a design consideration alongside productivity and aesthetics. To prevent COVID-19, especially when workers are returning to their respective workplaces, policymakers and building owners have to analyze, prepare, and implement various strategies for the safety of the workers. Based on the lessons learned and the vulnerabilities exposed during the pandemic stint of more than a year, a number of anti-COVID-19 considerations will be required for offices and workplaces for the safety of the workers. These requirements can be grouped under three broad headings: (1) regulatory environment, (2) engineering control environment, and (3) administrative and organization control environment, as shown in Table 2.

**Table 2.** Summary of office buildings and workplaces requirements to safeguard against COVID-19.

| Category | Subcategory | Requirements |
|---|---|---|
| Regulatory Environment | Agency Regulations | • Various international agency regulations that provide practical updated workplace guidelines to protect workplaces against COVID-19 [79]. |
| | Country Regulations | • COVID-19 guidelines differ across different countries, reflecting the differences in their health care facilities and strengths.<br>• In some instances, country-specific COVID-19 regulations may deviate from IHF regulations [114]. |
| | Operational Guidelines | • Constant updates of regulatory frameworks in workplaces are needed. Workplaces must be able to adapt their operational guidelines to encompass the latest regulatory updates at the national and international levels [78].<br>• Regulations should bring about behavioral changes required to protect workers from the COVID-19 pandemic [83].<br>• Specific behavior standards should be developed and implemented for each work sector based on general guidelines established by the health authority and with the participation of business chambers, trade unions, and government authorities [120].<br>• Workplace policies and protocols should be ready at the organization level to ensure employee safety and security during COVID-19 [79,121]. |
| Engineering Control Environment | Ventilation | • Increase fresh air supply and spent air exhaust rates in and out of workplaces through demand-controlled ventilation systems [56].<br>• The disabling of air recirculation systems should be strictly controlled [122].<br>• High-Efficiency Particulate Air (HEPA) filters must be installed to allow absolute clean air inflow [123].<br>• Control inflow rates of filtered air into workplaces to prevent virus-friendly environments (control indoor humidity and temperature) [33].<br>• Revamp industrial hygiene (IH) and occupational and environmental health and safety (OEHS) design to include considerations for virulent pathogens [88]. |
| | Distancing and Building Layout | • Designs must fundamentally consider health and sustainability [15].<br>• Working places should be more functional, healthier, and morally fulfilling [124].<br>• People in tall buildings may be more at risk [98].<br>• Install physical barriers in workplaces made of plexiglasses, polycarbonate, and tempered glass [72].<br>• Attention must be paid to avoid overcrowded interior spaces [125,126].<br>• Office spaces should become more varied, with fewer desks and more choices of spaces to meet, eat, exercise, and unwind [127].<br>• Opt for internal layouts with adequately open working spaces and allocate spaces for individuals [128]. |
| | Facilities and Equipment | • Provide personal digital terminals (tablets, laptops, desktop PCs, etc.) [33].<br>• Automate bathroom facilities (toilets, etc.) [129].<br>• Opt for devices with touch-free technology such as automated infrared temperature checks, facial recognition admittance devices, keycard swipers, and voice-activated elevators. [130].<br>• Consider selecting materials for surfaces (such as office desks, etc.) that sustain a shorter half-life of the SARS-CoV-2 virus [131]. |
| | Water and Sanitary Ware System | • During COVID-19 temporary building closure, routine flushing and shock disinfection were indicated as possible microbiological and legionella risk control methods [109]. |

**Table 2.** *Cont.*

| Category | Subcategory | Requirements |
|---|---|---|
| Administrative Control Environment | Workplace Safety Measures | • Together, source control and pathway control play an essential role in minimizing high-risk situations [132]<br>• Implement an efficient surveillance system that can track infection and transmission trends amongst employees and communication thereof to employees [133]. |
| | Cleaning and Disinfection | • Cleaning detergent and disinfectants are recommended to be made available at strategic places, especially at the frequent touch surfaces [134]. |
| | Emergency Plan Development | • Implement an efficient surveillance system that can track infection and transmission trends amongst employees and communicate to employees [133]. |
| | Business Continuity Guideline | • Establish teams of workplace experts that can enforce effective control measures [94].<br>• Allow for flexible working models, such as working from home [135]. |
| | Education and Communication | • Train workers to develop acquaintance with COVID-19 guidelines [39]. |

*6.1. Workplace Regulations*

The WHO's International Health Regulations (2005) (IHR) is an international agreement, agreed upon by 196 countries, with a scope to prevent, protect against, control, and provide a public health response to the international spread of disease. This regulation came into effect on 15 June 2007. The IHR addresses how different countries can take action against the spread of diseases such as COVID-19 or other public health issues. Therefore, in a sense, individual countries' COVID-19 health-protective guidelines must follow IHR regulations. Unfortunately, some countries were reported to have deviated from the IHR during COVID-19 [114].

Many international and national agencies, led by the WHO, have proactively issued COVID-19 safety guidelines for workplaces and are continually updating these guidelines as their understanding of the SARS-CoV-2 virus matures with time. The objectives of any of these agencies' guidelines are to provide national frameworks and organizational guidelines for the safe return of workers into workplaces. The authors of [136], reported that the COVID-19 guidelines of six countries during their initial outbreaks differed and reflected these nations' different healthcare strengths explicitly. The authors of [137] also pointed out the ad hocism of national and local COVID-19 guidelines across their 20-country study.

All regulations and guidelines, be they international or national, are fundamentally meant to prevent pandemic spread [138]. However, they may not address specific requirements for successful implementation in particular workplaces. For example, national or international guidelines may not specifically address how ventilation systems should work for the consideration of assembly line workers.

As different sectors of the economy reopen, countries need to develop sector-specific guidelines with collaboration between business chambers, workers' bodies, and government authorities so that benchmark standards can be implemented and followed by individual organizations [120]. Individual organizations should be ready to ensure employees' safety and security during COVID-19 [79,122]. Office managers should actively execute in-depth workplace analyses with expert input to develop the essential guidelines suitable for their workers. Constant updating of workplace guidelines according to the international and national guideline updates is required [78]. It must be noted that regulations cannot immediately bring about behavioral changes in workers [83]. Similarly, written policies or regulations alone cannot bring about the required behavioral changes [83]. Dennerlein et al., 2020 [139], therefore, indicated the necessity of multidimensional workplace regulatory frameworks that cover human factors and ergonomic principles, because, ultimately, employees must be cognizant of their responsibility towards their health.

## 6.2. Engineering Control Measure Environment

Efficient source control can play an important role in controlling infectious spreads in high-risk places [132], eliminating workers' risks of contracting pandemic diseases. Administrative actions that serve this purpose may include controlling office overcrowding, eliminating PPE doffing (such as masks, shields, etc.), and banning entry for visitors unwilling to maintain social distancing [140]. Source control may be especially difficult in sprawling office spaces, but surveillance systems may help by tracking and prohibiting possible protocol violations at their nascent stages. Bashir et al. (2020) [141] pointed out that even simple, low-cost end-to-end IoT architectures can aid in the surveillance of office environments to uphold standard operating procedures (SOP). With surveillance systems implemented, employers can ensure the strength of source control measures and quickly communicate about infection trends to inform employees of their actions [133].

Effective air cleaning, filtration, and ventilation systems are forms of engineering controls that can block virus entry pathways and maintain a healthy workplace environment [113]. The task of HVAC systems will be to prevent or eliminate airborne aerosols laden with micro respiratory droplets less than 5 microns in size from office environments. The epidemic task force of ASHRAE (American Society of Heating, Refrigerating and Air-Conditioning Engineers) suggested running office HVAC systems with at least minimum outdoor airflow rates to flush airborne viruses out of office spaces. HVACs should remain in operation until fresh air is replenished in every single employee's space within an office. Air cleaners and/or filters need to be utilized in conjunction with HVACs to prevent the recycling of unwanted aerosols back into the room [123]. ASHRAE also suggests using MERV (Minimum Efficiency Reporting Value) 13 or better rated air filters that can stop >95% of 0.30–1.0 micron-sized pathogens that are present in the air [56]. HVACs should also help maintain healthy workplace temperature and relative humidity (RH) that are unfriendly towards disease spread; in regard to this, research is still ongoing to establish the exact temperature and RH ranges that can best subdue the COVID-19 virus [142].

Health and sustainability should be at the core of any office building's architecture [15]. The COVID-19 pandemic made it starkly evident that designers and planners did not put enough importance on the issue of infectious disease control in building design. Before the pandemic, office buildings were developed with primary emphasis on productivity and aesthetics without sufficient consideration for infectious disease control features [143]. Therefore, office environments must become more versatile, infection resistant, and morally fulfilling for workers to achieve higher productivity [124]. Tall and crowded office buildings fitted with multiple fast-moving elevators are no longer looked upon proudly by employees due to their fear of virus infection. Instead, employees prefer lower-levelled office buildings with sprawling staircases [98]. Designers must also use building materials with a shorter half-life for the SARS-CoV-2 virus [131,144].

Open office layouts used to be much-touted for their productivity-enhancement capacity. However, their vulnerability to disease transmission was exposed during COVID-19, particularly by one case in Zurich where an entire team working in an open office became infected [87]. In India, workers wary of virus infection were reluctant to attend open-concept information technology call center offices. As such, designers must alter open plan layouts to make them more disease resilient [145]. Architectural innovations will be needed to create open office layouts that respect personal spaces and promote employee morale even during pandemics [146]. In addition, physical barriers made of plexiglass, polycarbonate, or tempered glass placed in refreshment rooms and meeting rooms can further reduce the possibility of close contact between employees [72].

Touch-free technologies can also mitigate the spread of virulent diseases in the office. Virus-laden surfaces, such as elevator keys, doorknobs, attendance registering keyboards, or even electrical switches, are prominent modes of disease transfer. Network-connected devices (Internet of Things, IoT networks) can help employees minimize physical contact with high-touch surfaces when passing through entry security systems, operating

elevators, accessing personal cabins, or using office equipment (computers, lights, fans, etc.) [130,147,148].

*6.3. Administrative and Organization Control Measure Environment*

Regardless of what sorts of management-enforced risk elimination and engineering controls are implemented in the workspace, they will be ineffective in keeping out COVID-19 unless efficient administrative follow-up is performed. Perhaps an organization's greatest asset against COVID-19 is motivated employees. Efficient administration can motivate employees to cooperate and maintain management-enacted guidelines and protocols for COVID-19 disease control. Without employee motivation, individual workers who defy protocols can undermine all management efforts in containing COVID-19 [149]. To this effect, the office administration's ability to involve workers in the safety planning and development of COVID-19 guidelines and protocols, such as workplace safety measures, cleaning and disinfection, emergency plan development, business continuity guidelines, and education and communication, will be immense. Administrators must not tire employees with rigorous COVID-19 protocol training but instead develop engaging and practical teaching sessions to prevent workers' inhibition against following the laid down protocols.

## 7. Conclusions

The current COVID-19 pandemic has brought different changes in our society. The continuance of the pandemic has forced humanity to think of erecting protective and preventive barriers that will allow people to return to everyday life. The rapid spread of COVID-19 forced the office and factory workers to leave their working premises and opt for work from home. However, offices and factories must function normally so as to maintain the economic health of any country. More resilient office and factory buildings with safety assurances are needed for workers to walk back to their offices and factories. The long COVID-19 stretch has exposed different workplace-related vulnerabilities that are required to be plugged in for workers to return to their workplaces. In this direction, this paper considered the vulnerabilities experienced in offices and working places under three major categories, the built environment, control environment, and regulatory environment, and then assessed methods to make workplaces conducive towards eliminating and controlling COVID-19 disease spread.

The COVID-19 pandemic made it clear that the built environment of offices was grossly incapable of controlling infectious disease spread as health and sustainability were not the fundamental part of the design aspects. Open space office layouts may provide cost efficiency as they enhance workers' productivity but can be an option during the COVID-19 pandemic. Office facilities and equipment must be equipped with more and more touch-free technologies. The control environment in offices needs to be reoriented, with the main focus on eliminating sources of infection and blocking the pathways of COVID-19 disease spread. Both engineering and administrative measures must work in a complementary mode to make the COVID-19 disease control environment fully effective in offices and factories. Regulations and guidelines enacted at the international or national level are essential in the context of the office environment to prevent pandemic spread. Agency regulations and country regulations are the important sources of updated research-based understanding that can be used for developing operational guidelines to protect workplaces against COVID-19. Specific behavior standards should be developed and implemented for each work sector based on general guidelines established by the health authority. The concerted efforts of business chambers, trade unions, and government regulations should bring about behavioral changes required to protect workers from the COVID-19 pandemic. Finally, the study suffers from a few limitations. The study entirely depended on the literature published for a limited period, starting from the origin of COVID-19. We found that number of research studies focusing on vulnerabilities of office

buildings due to COVID-19 diseases were comparatively less in number than residential building vulnerabilities.

**Supplementary Materials:** The following are available online at https://www.mdpi.com/article/10.3390/su132413636/s1, Table S1: Preventive and Protective Measures Against COVID-19 From International Agencies.

**Author Contributions:** Conceptualization, P.P., A.D. and O.C.; methodology, A.D. and O.C.; validation, O.C.; formal analysis, P.P. and A.D.; resources, P.P. and A.D.; data curation, O.C.; writing-original draft preparation, P.P. and A.D.; writing-review and editing, A.D. and O.C.; visualization, P.P.; project administration, P.P.; funding acquisition, P.P. and O.C. All authors have read and agreed to the published version of the manuscript.

**Funding:** This research was funded by "The 100th Anniversary Chulalongkorn University Fund for Doctoral Scholarship".

**Institutional Review Board Statement:** Not Applicable.

**Informed Consent Statement:** Not Applicable.

**Data Availability Statement:** Not Applicable.

**Acknowledgments:** This research was (partially) supported by the Ratchadapisek Sompoch Endowment Fund (2021), Chulalongkorn University (764002-ENV). We acknowledge the funding support provided by the Second Century Fund (C2F) from Chulalongkorn University awarded to Abhishek Dutta's postdoctoral fellowship.

**Conflicts of Interest:** The authors declare no conflict of interest.

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
