# Peer review of "COVID-19 Experience Transforming the Protective Environment of Office Buildings and Spaces"

_sustainability, doi:10.3390/su132413636_

Round 1
Reviewer 1 Report
- The abstract should highlight the innovative aspect of the study compared to previous studies.
- The literature is well structured and complete. Please add some updated literature. I offer a few references below and I think it will be useful to add them ;
DoÄŸan, B. Mehdi Ben Jebli, Khurram Shahzad, Taimoor Hassan Farooq, Umer Shahzad, (2020). Investigating the Effects of Meteorological Parameters on COVID-19: Case Study of New Jersey, United States, Environmental Research, 191, 110148, https://doi.org/10.1016/j.envres.2020.110148.
Khurram Shahzad, Taimoor Hassan Farooq, Buhari DoÄŸan, Li Zhong Hu & Umer Shahzad (2021) Does environmental quality and weather induce COVID-19: Case study of Istanbul, Turkey, Environmental Forensics, DOI: 10.1080/15275922.2021.1940380
- The databases are informative and adequate
- The sources of data need to be cited properly in the paper – every reader should understand how to access these.
- The research limitations should be stated clearly.
- Policy implications of the results are weak and should be better articulated.
Reviewer 2 Report
Aim and objectives of the study
Recommendation to restructure the inclusion study aim and objective in a fresh or new section heading (refer to line 145-154). Or
To consider to re-title/name the Section 2 as- Problem Statement: Workplace Regulation to Mitigate Covid 19, this will enable the author to conclude the problem statement section with a set of objectives as outlined, refer to line 145-154.
The article needs to be structured according to research questions that define the 4 sets of objectives that the author intends to explore theoretically.
Authors need to describe the methodology used in the paper i.e. the literature review process answering the research questions developed based on the problem statement which has been highlighted in Section2. Then align with the overall findings against the 4 sets of objectives.
Agostinho (2020) [72] pointed out … safe working distance between one another. Line 269 - author discussion on safe distance is limited to 6 feet, please refer and include the following work whereby current research has measures the effects of 3 feet's safe distance:-
Effectiveness of 3 Versus 6 ft of Physical Distancing for Controlling Spread of Coronavirus Disease 2019 Among Primary and Secondary Students and Staff: A Retrospective, Statewide Cohort Study
Clinical Infectious Diseases, Volume 73, Issue 10, 15 November 2021, Pages 1871–1878, https://doi.org/10.1093/cid/ciab230
Apply a suitable referencing/citation format consistent throughout the article.
Samples:-
Honey-Roses et al. (2020) [19] concluded …
Tokazhanov et al. (2020) [22] noted
Jiang et al. (2021) [23] pointed out
The 121
Architectural Society of China (2020) [24]
Awada et al. (2021) [25] confirmed
Grammatical (Sampling)
The article needs to be proofread, following are the samples:-
Since workers can come into close contact with colleagues and visitors from all (Line 101) different localities and of various COVID-19 pandemic infectivity levels, virus exposure (line102) can occur at any time within the workplace [31].
These questions will all be addressed in the currently timely context of (line153) supporting a smooth transition from 'work from home' to 'work from the office' as usual.
Both governments and industrial (line110) associations must take great care in devising industry-specific risk and safety guidelines (line111) for workers to abide by.
Experts are (line129) apprehensive of lost productivity as working from home has distanced workers from (line130) their coworkers and managers [27].
grossly overlooked the issue of infectious disease control in building (line577) designs.
COVID-19 has shown the need for built office environments and spaces to be (line481) unfriendly against COVID-19 type viruses [15].
with only a glancing consideration (line579).
Designers must also opt for building materials that sustain a (line585) shorter half-life for the SARS-CoV-2 virus [131], [143].
Line 20-21
lack of safety feeling, and workers switch over to a safer work from home system.
Line 22
to consider changes in the three broad environmental dimensions to shed their vulnerability status experienced during the pandemic
Therefore, office environments (*line580) must become more versatile, infection-resistant, and morally fulfilling for workers to (line581) achieve higher efficiency [123].
Section 6 can be removed and related facts can be restructured into the text body, as the author described earlier in the previous section.
The aims and objectives outlined in the earlier part of the article do not require any aspect of Thailand. The conclusion and recommendation are conceptual, reflecting how countries adopt necessary control measures.
The paper is very long and can be kept simple and concise into less than 20 pages maximum if possible. However, the referencing (more than hundreds) made is robust.
Round 2
Reviewer 2 Report
The authors' have made necessary amendments as commented.